# Saccadic suppression as a perceptual consequence of efficient sensorimotor estimation

Frédéric Crevecoeur[1,2]*, Konrad P Kording[3]*

[1]Institute of Information and Communication Technologies, Electronics and Applied Mathematics, Université catholique de Louvain, Louvain-la-Neuve, Belgium; [2]Institute of Neuroscience, Université catholique de Louvain, Louvain-la-Neuve, Belgium; [3]Rehabilitation Institute of Chicago, Northwestern University, Chicago, United States

**Abstract** Humans perform saccadic eye movements two to three times per second. When doing so, the nervous system strongly suppresses sensory feedback for extended periods of time in comparison to movement time. Why does the brain discard so much visual information? Here we suggest that perceptual suppression may arise from efficient sensorimotor computations, assuming that perception and control are fundamentally linked. More precisely, we show theoretically that a Bayesian estimator should reduce the weight of sensory information around the time of saccades, as a result of signal dependent noise and of sensorimotor delays. Such reduction parallels the behavioral suppression occurring prior to and during saccades, and the reduction in neural responses to visual stimuli observed across the visual hierarchy. We suggest that saccadic suppression originates from efficient sensorimotor processing, indicating that the brain shares neural resources for perception and control.

**\*For correspondence:** frederic.
crevecoeur@uclouvain.be (FC);
koerding@gmail.com (KPK)

**Competing interests:** The
authors declare that no
competing interests exist.

**Reviewing editor:** Emilio
Salinas, Wake Forest School of
Medicine, United States

## Introduction

People skillfully combine acquired knowledge, and sensory feedback, a combination that is typically modeled using Bayesian statistics (*Körding, 2007*; *Angelaki et al., 2009*). This framework effectively captures behavior in numerous tasks broadly corresponding to perceptual decision-making (*Ernst and Banks, 2002*; *van Beers et al., 1999*; *Fetsch et al., 2011*; *Drugowitsch et al., 2014*; *Acuna et al., 2015*), or online movement control (*Wolpert et al., 1995*; *Körding and Wolpert, 2004*; *Izawa and Shadmehr, 2008*; *Crevecoeur et al., 2016*). Although perceptual decision-making and sensorimotor control are often considered different phenomena, they cannot really be dissociated in the real world – we need to use the same brain for movement and perception (*Cisek, 2012*; *Wolpert and Landy, 2012*). Perceptual decision-making and sensorimotor behaviors may thus be linked.

A salient case of crosstalk between perception and sensorimotor behavior is *saccadic suppression*: visual acuity is reduced around the time of a saccade. It is often assumed that this mechanism maintains stable perception of our surroundings (*Wurtz, 2008*). However, the behavioral and neural dynamics of saccadic suppression are difficult to explain if it were purely related to compensating for shifts in the retinal image induced by saccades and by the need to maintain perceptual stability.

Indeed, previous work has shown that saccadic suppression is controlled centrally, and typically lasts for >100 ms even for saccadic movements of ~50 ms (*Ibbotson and Krekelberg, 2011*). Furthermore, simulating the displacement of the retinal image without a saccade does not elicit similar suppression as during real saccades (*Diamond et al., 2000*; *Thiele et al., 2002*). As well, the

**eLife digest** Although we have the impression that our eyes move smoothly from place to place, we in fact perform rapid eye movements called saccades several times per second. Experiments have shown that our ability to perceive contrast and flashes decreases before and during each saccade. This phenomenon is known as saccadic suppression.

A prevailing hypothesis to explain saccadic suppression suggests that by making vision temporarily less sharp for the rapid eye movement, the nervous system discards visual information about movement and helps us to perceive the world as stable. However, this does not explain the timing of saccadic suppression. Indeed, for saccades of about 50 milliseconds, the brain begins to reduce the sharpness of vision roughly 100 milliseconds before each eye movement begins. Why does the brain discard so much visual input?

To answer this question, Crevecoeur and Kording generated a computer model that took into account three properties that previous experiments have detected in animal nervous systems. First, transferring information between the retina and the neurons that control the movement of the eyes involves delays. Second, when neurons generate commands to move the eyes, they also show random fluctuations in activity that increase with the intensity of the commands. And third, visual information can still influence eye movement during a saccade. As a result of incorporating these three properties, the model predicted optimal timings for saccadic suppression that correspond to those that occur in real life.

Visual perception and the control of eye movements have often been considered as separate functions of the brain. However, the model generated by Crevecoeur and Kording suggests that perception and the control of eye movement may in fact involve common brain regions. Further research is now needed to investigate predictions made by the model, which should provide new insights into how the brain supports vision.

reduction of visual acuity was reported to selectively impact the magnocellular pathway (*Burr et al., 1994*), although motion detection is still active (*Castet and Masson, 2000*). It is unclear why maintaining perceptual stability would require such a long, powerful, and selective suppression of sensory feedback, if it were purely related to perception, and independent of motor control. After all, there is a price to be paid to discard so much sensory information for some 100 ms. Thus, saccadic suppression is a complex phenomenon, for which a meaningful function has not been clearly identified.

Here we phrase this problem in the framework of Bayesian estimation during closed-loop control of saccades. In this framework, we show theoretically that the timing of saccadic suppression is expected if the brain uses the same posterior beliefs about the state of the eye for perception and control. Indeed, our model shows that uncertainty about the instantaneous state of the eye should increase with motor commands as a result of signal-dependent noise and of sensorimotor delays, making delayed sensory information less reliable around the time of movement. In an optimal estimation framework, this gives lower weights to sensory inputs when we move. Our study thus shows how sensorimotor control can give rise to sensory suppression in an efficient brain, provided that the nervous system uses a common substrate for perception and for control. We discuss how this theoretical result may arise from shared neural resources supporting perceptual and motor systems.

## Results

### An optimal control model of eye movements

If we want to explore the relationship between saccadic suppression and control we need to model the underlying system. First, the nature of the representation matters: although saccades are often simplistically viewed as ballistic (or *open-loop*) movements, these movements are monitored online through the corollary discharge (*Van Gisbergen et al., 1981*; *West et al., 2009*; *Goossens and Van Opstal, 2000*; *Xu-Wilson et al., 2011*; *Sommer and Wurtz, 2008*; *Optican, 2009*). Second, sensory feedback matters: we are not 'blind' during saccades. There is no peripheral interruption of sensory

inflow, and information about specific spatiotemporal frequency or color is still good (*Burr et al., 1994*; *Burr and Morrone, 1996*). Moreover, target jumps during long saccades can influence movement (*Gaveau et al., 2003*). We should thus model saccades as driven by closed-loop control (*Figure 1a*).

To describe saccades in the context of closed loop control, we model a controller that takes the sensory feedback and the corollary discharge as input, and outputs motor commands. We employ a Linear-Quadratic-Gaussian (LQG) controller, which can deal with noise both in sensory feedback and control signals. We use a second order model for the oculomotor plant (see Materials and methods). This explicit model of saccadic control allows us to derive predictions of eye movement behaviors and gives us a control process that we can relate to saccadic suppression.

The important feature of this control design in the context of this paper is its state estimator. The control of saccadic eye movements relies on the corollary discharges as well as on sensory feedback, which jointly allow state estimation. This state estimator has two main components. The first is a forward model that dynamically updates the current estimate based on the corollary discharge (*Figure 1a*, bottom: Forward Model). The output is a prior estimate of the next state at the next step. The second component is the sensory extrapolation, which combines the delayed sensory feedback with the corollary discharge to estimate the current state (*Figure 1a*: Sensory Extrapolation, red). This sensory extrapolation is critical for the behavior of the model.

The presence of sensory extrapolation is supported by previous studies showing that error signals used to generate saccades depend on an estimate of the present state of the eye or of the target (*Bennett et al., 2007*; *Ferrera and Barborica, 2010*; *Diaz et al., 2013*; *Blohm et al., 2005*; *de Brouwer et al., 2002*), which clearly requires extrapolation of sensory feedback. Indeed, because the system only has access to the delayed feedback, this feedback must be extrapolated to compare it with the one step prediction, or prior. This operation does not appear explicitly in standard control models in which sensorimotor delays were considered (*Izawa and Shadmehr, 2008*; *Crevecoeur et al., 2016*; *Todorov and Jordan, 2002*; *Crevecoeur and Scott, 2013*), because these previous studies used system augmentation, and the sensory extrapolation in this case falls out of the block-structure of the model. However, this component is necessary, and

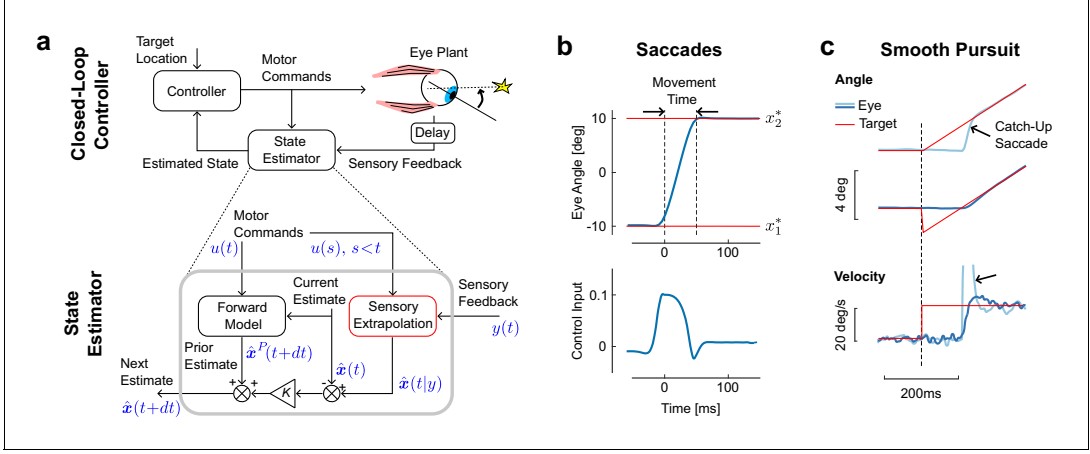

**Figure 1.** Model architecture and simulations of eye movements. (**a**) Schematic representation of the control and estimation architectures. We consider a closed loop controller based on optimal feedback control and state estimation. The dynamics of the eye plant corresponded to a second order system with time constants taken from the literature (13 ms and 224 ms). Bottom: Optimal state estimator based on usual Kalman filtering, and augmented with the extrapolation of sensory feedback to compensate for sensorimotor delays (Sensory Extrapolation, red box). The symbolic representation of the signals in blue follows the same notations as in the Materials and methods: $y(t)$ is the sensory feedback, $\hat{x}(t|y)$ is the extrapolation of sensory feedback, $u(.)$ is the sequence of previous and current control commands, $\hat{x}^P(.)$ and $\hat{x}(.)$ are the prior and posterior estimates at the corresponding time steps. (**b**) Top: Modeled saccadic eye movement from the first ($x_1^*$) to the second fixation target ($x_2^*$). Bottom: Associated control function. Time zero corresponds to the end of the fixation period to the first target. (**c**) Illustration of the sensory extrapolation performed in the state estimator. The simulated task is to track the target, which suddenly starts moving (velocity jump) with or without position jump in the opposite direction. The simulated eye trajectory shows how the extrapolation of target motion over the delay interval generates a catch up saccade (black arrow). This compensatory movement is also illustrated in the velocity trace.

ignoring it can lead to instability (*Crevecoeur and Scott, 2013*). In the present model, the sensory extrapolation is performed explicitly (*Equation 5*), which also allows us to incorporate the impact of the signal-dependent noise that accumulates over the delay interval during the extrapolation (see also Materials and methods). The model then corrects the one step prediction, weighting the difference between feedback and expected feedback optimally (*Figure 1a*, $K(t)$ is the Kalman gain).

## Saccades and smooth pursuit

The first behavior that our model must describe is a saccade. The model reproduces stereotyped, step-like trajectories (*Figure 1b*, top), like those found during real saccades. Moreover, the associated commands provide a typically wide agonist burst, followed by a short, sharp antagonistic inflection, which stabilizes the eye at the target (*Figure 1b*, bottom). This pattern of control, shaped by the fast time constants of the oculomotor plant, is compatible with the pattern of burst neurons that generate saccades (*Van Gisbergen et al., 1981*). Thus the model replicates both behavioral and physiological aspects of saccadic eye movements.

A second behavior that our model can capture is smooth pursuit. We do not imply that these two behaviors are supported by the same neural hardware, and the model does not make any prediction about their neural implementation. Instead, we simply assume that optimal state estimation underlies both saccades and pursuit, which is in agreement with the hypothesis that these movements are distinct outputs of shared sensorimotor computations (*Orban de Xivry and Lefèvre, 2007*; *Krauzlis, 2004*). The model reproduces typical responses to changes in target velocity, occurring with or without initial target jump (*Figure 1c*). When the target starts moving (velocity jump), position error accumulates over the delay interval, which in turn requires a rapid compensatory movement to catch up with the target (*Figure 1b*, light blue). Although the controller was not explicitly designed to model the interaction between saccades and pursuit, the catch-up saccade in *Figure 1b* naturally falls out of the simultaneous correction for errors both in position and velocity. In contrast, when the target jumps backwards at the onset of the velocity jump (*Figure 1c*, dark blue), the eye starts moving smoothly and there is no catch-up saccade (*Rashbass, 1961*). The model also reproduces corrections following perturbations applied during movement through internal monitoring of the corollary discharge, as well as online corrections for target jumps occurring during long saccades (simulations not shown). In all, the model generates typical trajectories and control commands associated with eye movements, and reproduces the dynamic estimation of the target resulting from the sensory extrapolation.

## Saccadic suppression as a consequence of optimal estimation

Our muscles produce signal dependent noise; the stronger the muscles pull, the more noisy the state. The phasic activity associated with the agonist burst induces a peak in the variance of the control signal (*Figure 2a*, solid). Thus motor commands produce instantaneous noise, and because of the delay, there is no way for the nervous system to directly subtract or filter out this noise. As a consequence, the extrapolation error computed over an interval that includes even a fraction of the control burst has higher variance. In other words, moving the eye effectively induces visual uncertainty, which can only go back to baseline after the end of muscle activation.

The time-varying variance induced by control-dependent noise has a direct impact on the weight of retinal signals, through the Kalman gains ($K(t)$, *Figure 2b*). Recall that this matrix weights the difference between the current and expected estimates of both position and velocity, conditional upon the available visual information (see Materials and methods, $y(t)$, *Equation 4* and *Figure 1*) to correct the one-step prediction. We thus focus on the weight of position feedback, as it appears closely related to saccadic suppression.

During saccades, the extrapolation variance increases as a result of signal dependent noise and of sensorimotor delays ($V_t$, *Equation 8*). As the Kalman gain is inversely proportional to this variance, the transient increase associated with the agonist burst generates a reduction in $K(t)$, and thus lowers the weight of position feedback about the eye position in the state estimator. And indeed, sensory suppression is seen before and during the time of simulated saccades (*Figure 2b*). The period of suppression predicted by the model is long because the high variance period includes the movement time in addition to the delay (gray rectangle in *Figure 2c*). The model predicts that the onset

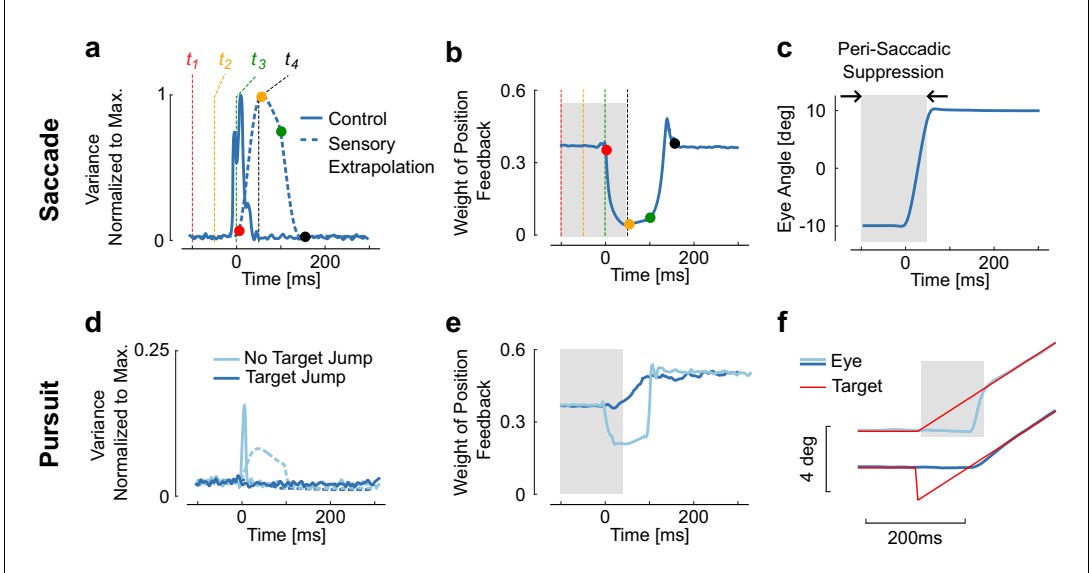

**Figure 2.** Reduction of sensory weight. (a) Variance of the control signal (solid) and of the extrapolation of sensory feedback (dashed). Four times are represented to illustrate how signal-dependent noise impacts the extrapolation of sensory feedback is ($t_1 = -100$ ms, $t_2 = -50$ ms, $t_3 = 0$ ms, and $t_4 = 50$ ms). The dots with similar color represent the moment when the information at the corresponding time is available ($t_i + 100ms$). Observe the increase in extrapolation variance associated with stimuli between $t_1$ and $t_2$. (b) Weight of the position feedback for correcting the estimate of the position. This weight is directly taken from the Kalman gain matrix. The reduction in Kalman gain at each selected time point is directly linked to the increase in variance. (c) Representation of a modeled saccadic eye movement, with the gray area corresponding to the interval of time during which sensory input is given less weight as a result of the extrapolation variance. (d) Control and extrapolation variance normalized to the maximum values obtained for saccades of 20 deg (top traces). (e) Weight of sensory feedback for the two simulations. Observe that although the catch-up saccade is very small (~2 deg), the transient increase in extrapolation variance gives rise to a reduction in weight. (f) Illustration of the smooth pursuit task with (dark blue) or without (light blue) initial target jump occurring simultaneously with the velocity jump. The absence of target jump evokes a catch-up saccade, which is associated with ta reduction in sensory weight. There is no reduction with the initiation of smooth pursuit.

of saccadic suppression should precede movement onset by a time interval equal to the delay, which was fixed to 100 ms in the model (see Materials and methods), although previous work suggested that it could be shorter (*Gaveau et al., 2003*). Considering that processing times in the retina approach ~50 ms (*White et al., 2009*), a conservative estimate for the onset of suppression according to the model ranges from 100 ms to 50 ms prior to movement onset. Our theoretical predictions thus indicate that feedback about retinal stimuli from this time window should be given less weight to optimally estimate the state of the eye.

We can also see related effects in the simulated pursuit task. The presence of a catch up saccade, even a small one (4deg in *Figure 2*), is sufficient to evoke a transient increase in extrapolation variance (*Figure 2d–f*). This results in a reduction in the weight of sensory feedback with similar timing as for larger saccades. In contrast, when the eye starts moving smoothly (*Figure 2f*), there is no catch-up saccade needed and the model predicts no visible change in the weight of sensory feedback. Because the model is linear, there is no transition between the simulated pursuit and saccade task, thus the apparent transition in the behavior results from the correction for the error in position that accumulates during the delay interval (*Figures 1* and *2*, light blue). The fact that suppression occurs for saccades specifically results from the high control signals required for these movements. In contrast, the pursuit task without a saccade uses smaller control signals that do not evoke any visible change in the weight of sensory feedback. The behavioral finding (*Schütz et al., 2007*) that saccades but not smooth pursuit elicit suppression of sensory feedback, and that the suppression scales with the amplitude of the catch-up saccade, directly results from this model.

Behaviorally, we can analyze data from perception experiments. According to our hypothesis, suppression should occur prior to movement onset, reach a maximum close to movement onset (*Figure 3b*), and scale with the movement amplitude with relatively invariant timing across amplitudes. Interestingly, this goes even down to the level of microsaccades, inducing partial suppression

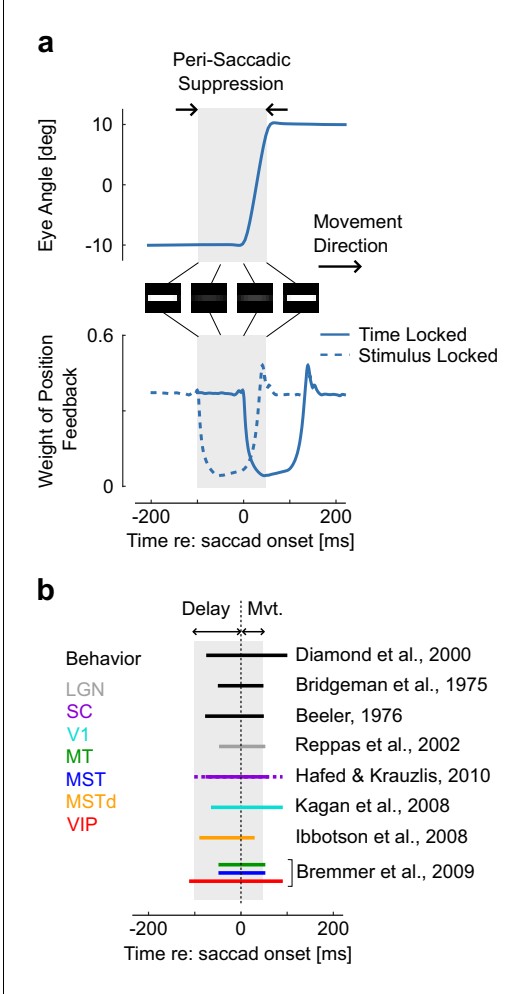

**Figure 3.** Representation of a simulated 20 deg saccade ad dynamic weight of sensory feedback, with the perisaccadic suppression highlighted in gray. These traces are similar as *Figure 3a and c*. The images represent the convolution of a horizontal stripe with a Gaussian kernel with variance proportional to the extrapolation variance to highlight that assigning higher variance may lead to reduced contrast, even when the movement is aligned with the stimulus orientation. Times correspond to the *Figure 3*. The decrease in Kalman gain occurs from 0 to 150 ms relative to saccade onset (solid trace: time locked), thus the window during which stimuli are suppressed corresponds to −100 to 50 ms (dashed trace: stimulus-locked). (b) Illustration of how the predicted saccadic suppression compares with previously reported suppression from behavioral and neural data. The duration of the perisaccadic suppression in the model is the sum of the temporal delay and of the movement time as represented above with the gray rectangle. Comparisons are approximate as movement time was not the same across all studies. The solid and dashed traces for saccadic suppression in SC indicates the range of onset and offset as given by *Hafed and Krauzlis (2010)*. Other intervals of saccadic

*Figure 3 continued on next page*

despite being very small in amplitude (*Hafed and Krauzlis, 2010*). These known properties of saccadic suppression are in line with the model prediction (*Figure 3b*, black): contrast sensitivity is reduced and visual stimuli such as flashes, gratings, or small displacements are less likely to be accurately perceived (*Diamond et al., 2000*; *Burr et al., 1994*; *Watson and Krekelberg, 2011*; *Burr et al., 1999*; *Bridgeman et al., 1975*; *Beeler, 1967*). This even happens when the stimuli are chosen so that the eye movement does not change the retinal image, which is compatible with the model (see the simulated contrast reduction of a white stripe, *Figure 3a*). Finally, although timing is preserved across amplitudes (*Ibbotson and Krekelberg, 2011*), the model predicts that the magnitude of suppression scales with the saccade amplitude as observed experimentally (*Ridder and Tomlinson, 1997*). This scaling is a direct consequence of signal-dependent noise.

A brief increase in the Kalman gain following the saccade can be observed in *Figures 2* and *3*. This increase is due to the fact that the absolute value of the motor command is transiently lesser than the activity required to maintain the eye at the eccentric target. Thus, during this short interval, the Kalman gain becomes greater than during the simulated fixation. This feature resembles post-saccadic enhancement, which characterizes the enhanced motor response to a velocity jump in the target following a saccade (*Lisberger, 1998*; *Ibbotson et al., 2007*). However, this transient increase in the model was not sufficient to generate a behavior similar to the one observed experimentally, despite the fact that larger differences in the Kalman gain can do so (simulations not shown). This observation and the fact that motion detection is active during saccades (*Castet and Masson, 2000*) suggest that the model needs further refinement to fully capture the processing of velocity signals during and after saccades.

The model also predicts changes in neural activity relative to the timing of saccadic suppression. One way to implement the Kalman gains is to simply drive neurons less strongly when there is more uncertainty. This should predict reduced firing rates around the time of saccades. Indeed, a large number of experimental studies have found such a neural suppression across the hierarchy of visual areas. (*Ibbotson et al., 2008*). The pathways begin with the lateral geniculate nucleus (LGN) (*Reppas et al., 2002*) and the superior colliculus (SC) (*Hafed and Krauzlis, 2010*), and continue in

*Figure 3 continued*

suppression were drawn following the authors' summary or based on visual inspection of the corresponding references.

the cortical areas V1 (*Kagan et al., 2008*), MT, MST, MSTd (*Ibbotson et al., 2008*; *Bremmer et al., 2009*), and VIP (*Bremmer et al., 2009*) (*Figure 3b*, colored bars). Interestingly, the timing is very similar across brain regions, which emerges naturally from the fact that the loop through the outside world with its delays is the dominating timescale. Thus there is suppression of visual signals across the entire visual hierarchy consistent with a sensorimotor origin of saccadic suppression.

## Discussion

We have presented a feedback control model that assumes signal-dependent noise and delays, and uses state estimation to optimally control eye movements. It is built on the insight that motor noise is unavoidable and produces sensorimotor uncertainty. It is also based on the key assumption that saccades are supported by closed-loop control including retinal signals. This assumption is further developed below. This model allowed us to propose the hypothesis that saccadic suppression originates from efficient sensorimotor integration. Behaviorally, it describes the dynamics of both smooth pursuit and saccades. Perceptually, it describes the suppression of sensation around the time of saccades. Neurally, it captures the reduction of neural responses to visual stimuli presented before or during saccades.

The important motivation behind this study was that cancelling the retinal shift induced by the saccade, as commonly assumed, does not explain the phenomenon of saccadic suppression. Indeed, suppression in this case should only occur when the eyes move, and should not be stronger than the moderate loss in performance associated with saccades simulated as a rapid displacement of the visual scene (*Diamond et al., 2000*). All discarded information beyond movement-related effects would otherwise represent a net loss (up to ~100 ms for some brain areas, *Figure 3b*). Thus it is clear that saccadic suppression is either very inefficient, or that maintaining a stable visual scene is just not its only purpose. We provide an alternative hypothesis that captures suppression qualitatively in the context of sensorimotor control. Rather than providing a definite answer to why suppression occurs, we highlight a plausible explanation and expect that it provide an insightful framework for interpreting data about visual processing.

Although the model is not straightforward to test experimentally, our assumption about a common origin for saccadic suppression and movement control makes testable predictions for prospective experimental work. For instance, as we suggest that perception is impaired by sensorimotor control, it is conceivable that control might be impaired by a perceptual task. That is, if it were possible to train participants to pay attention to visual stimuli displayed during saccades, thereby increasing the weight of sensory feedback, then the theory predicts that movement trajectories should become more variable as a result of suboptimal state estimation. There is already clear evidence that the locus of attention and the goal of saccadic movements are linked (*Kowler et al., 1995*) (and many references thereto). Here our specific prediction is that saccade trajectories should become more variable from trial to trial when participants are forced to use sensory information presented during the interval of saccadic suppression. As well, assuming that saccadic suppression is directly linked to the variance of sensory feedback through the Kalman filter, the model predicts that varying the reliability of sensory information may have an impact on the magnitude of saccadic suppression. Observe that these two predictions also assume that suppression can be flexibly modulated dependent on the behavioral context, which to our knowledge has not been documented.

In addition to capturing the major aspects of behavioral and neural suppression, our model explains the previous findings of Watson and colleagues (*Watson and Krekelberg, 2011*), who investigated the detection of noisy gratings in humans, and found that the best explanation for saccadic suppression was a stimulus-independent reduction in the response gain. This result is a key aspect of saccadic suppression: the retinal images do not become intrinsically noisier; instead it is the visual system that responds less to a given stimulus. Our model also accounts for this observation: by reducing the sensory weight in the Kalman gain, the controller becomes less sensitive to sensory information. This is due to the uncertainty induced by the motor commands, which is clearly

independent of the retinal image. The contribution of our model is to show that such stimulus-independent reduction in response gain may be rooted in efficient computations about the state of the eye.

We propose this mechanism as a plausible origin of saccadic suppression, but cannot indicate how the visual system performs this operation at the level of neural circuits. We draw a qualitative link between Kalman filtering and the reduction in sensory weight or neural excitability, and thus this link remains speculative. However, the model does provide hints about what to look for. First, the increase in extrapolation variance clearly results from convolving the motor-dependent noise with the expected eye dynamics over the delay interval. Second, this increase is directly proportional to the integrated motor command. Thus, convolutional networks in the visual system receiving the corollary discharge as input may easily implement a reduction in the gain of neural responses that achieves statistically optimal sensory weighting. Any anatomical or functional similarity between these putative neural operations and neural data may thus provide insight into the circuitry underlying state estimation.

A compelling aspect of our model is its simplicity, as the distinct behaviors and the dynamic estimation simply fell out of the simplest instance of linear stochastic optimal control (LQG). Besides saccadic suppression, our model succeeded at the difficult task of controlling fast movements with comparatively long delays, without artificially interrupting the sensory inflow. While previous models of saccadic control tend to only consider *open-loop* controllers (*Harris and Wolpert, 1998*), or closed-loop control with internal feedback only (*Optican, 2009*; *Jürgens et al., 1981*; *Chen-Harris et al., 2008*), there is evidence that sensory information remains available and may influence online control. Indeed, motion detection during saccades is not suppressed (*Castet and Masson, 2000*), and peri-saccadic target jumps evoke adaptation (*Panouillères et al., 2016*). In addition, a large retinal slip prior to saccade initiation can elicit curved movements (*Schreiber et al., 2006*), indicating that retinal information prior to saccade onset can influence online control. Finally, Gaveau and colleagues reported partial corrections of eye trajectories following target jumps occurring during long saccades (*Gaveau et al., 2003*). The fact that these corrections accounted for a small proportion of the target jump can be explained in the model, as a lower weight of sensory feedback leads to only partial correction of the target jump (estimates take longer to converge to the true value). Thus, although evidence may not be definitive, these previous observations collectively suggest that sensory feedback must be considered in a model of neural control of saccades.

Based on this assumption, our model predicts that sensory feedback must be strongly reduced, but not completely suppressed, as observed experimentally (*Castet and Masson, 2000*). This is because the Kalman filter achieves an optimal projection in the probabilistic sense, by making the estimation error orthogonal to (or statistically uncorrelated with) the estimated state. Thus, the decrease in the Kalman gain during movement indicates that state information prior to the saccade still carries some information about the current state, and thus can be exploited to derive optimal estimates. The resulting control law (see Materials and methods, *Equation 10*) plays the role of a burst generator, and can be easily inserted as such in more complex models of gaze control.

We have formulated the hypothesis that saccadic suppression originates from sensorimotor processing, although suppression has been characterized behaviorally as a perceptual phenomenon. Thus our theoretical developments imply that perception and control share a common neural substrate in the visual system. There are already strong pieces of evidence for shared resources. Indeed, previous work emphasized that perception and action share estimates of target speed (*Priebe and Lisberger, 2004*). Recently, a strong link between saccadic suppression and visual-motor neurons has been established in superior colliculus of macaque monkeys (*Chen and Hafed, 2017*). Furthermore, the motion on its own must not be suppressed to maintain perceptual stability, instead it must be equal to the commanded movement monitored online, thus perception is also conditional upon the ability to integrate extra-retinal signals accurately, both during saccades and pursuit (*Sommer and Wurtz, 2008*; *Blohm et al., 2005*; *Hafed and Krauzlis, 2010*; *Bedell and Lott, 1996*).

If perceptual and sensorimotor processes were completely decoupled, posterior beliefs about sensory information could be separated from movement-related effects, and perception around the time of saccades could be as good as during simulated saccades (*Figure 4*, H1). Alternatively, a motor origin of saccadic suppression implies that the same posterior beliefs are shared for perception and control, which is suboptimal as it impacts perception of otherwise reliable sensory signals (*Figure 4*, H2). Thus the hypothesis of shared resources requires a functional explanation. Although

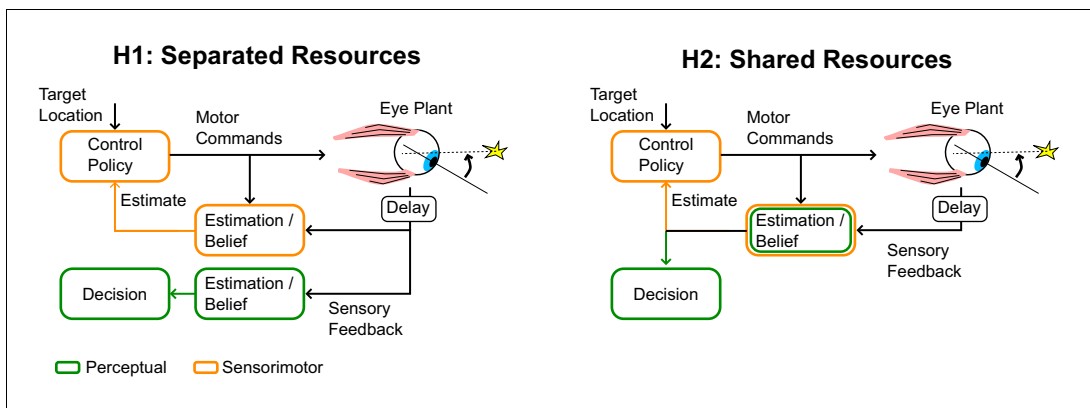

**Figure 4.** Schematic illustration of separate or shared resources hypotehses. In the hypothesis of separated resources (H1), computations of the posterior belief are carried out independently for perception and control. In this scenario, the uncertainty induced by the control commands does not impact the perceptual estimate. This possible architecture is optimal in the sense that it would minimize loss of sensory information. In the hypothesis of shared resource (H2), the computation of the posterior belief about the state of a variable is shared for perception and control, thus both processes are similarly influenced by control-dependent noise. Although the first hypothesis is optimal, the second hypothesis is more efficient in terms of neural resources, and is also self-consistent (see Discussion).

perception on its own is suboptimal during saccades (we discard a lot of meaningful information), the shared resources model is clearly cheaper in terms of neural resources. It may thus be globally optimal to tolerate perceptual loss during saccades rather than commit to more neural resources for visual processing, considering perceptual and control systems together. Using the same posterior belief for perception and action also ensures self-consistency, in the sense that the same stimulus is not deemed more or less reliable dependent on how we use it. Self-consistency is known to characterize perceptual judgment tasks, where participants make continuous use of the hypothesis to which they previously committed (*Stocker and Simoncelli, 2007*). Our model suggests that similar principles may govern the use of posterior beliefs about the state of the eye for perception and control, indicating that these functions emerge from a shared neural substrate. We hope that future work will investigate whether this theory is generally applicable to other examples of active sensory suppression associated with voluntary actions such as force generation and reaching (*Chapman et al., 1987*; *Blakemore et al., 1999*; *Seki et al., 2003*).

## Materials and methods

### Biomechanical model

We consider a second-order, low-pass filter as a biomechanical model of the oculomotor plant. Based on previous modeling work (*Robinson et al., 1986*) we set the time constants to $\tau_1 = 224ms$ and $\tau_2 = 13ms$. In the sequel, scalars are represented with lower-case characters, vectors with bold lower-case and matrices with capitals. Thus the state-space representation of the continuous-time differential equation representing the eye dynamics was:

$$\begin{bmatrix} \dot{x}_1 \\ \dot{x}_2 \end{bmatrix} = \begin{bmatrix} 0 & 1 \\ -1/(\tau_1\tau_2) & -(\tau_1+\tau_2)/(\tau_1\tau_2) \end{bmatrix} \begin{bmatrix} x_1 \\ x_2 \end{bmatrix} + \begin{bmatrix} 0 \\ 1/(\tau_1\tau_2) \end{bmatrix} u \tag{1}$$

where $x_1$ is the eye angle, $x_2$ is the eye velocity, $u$ is the command input and the dot operator is the time derivative. The explicit dependency on time was omitted for clarity. This representation takes the form

$$\dot{\mathbf{x}} = A\mathbf{x} + Bu \tag{2}$$

with $\mathbf{x} := [x_1 \ x_2]^T$ representing the state of the system.

The plant model was then augmented with the target position and target velocity, and transformed into discrete time model to include sensorimotor noise. The discrete-time stochastic dynamics governing the change of state over time, and the equation describing the visual signals available in the brain ($\mathbf{y}(t)$) are as follows:

$$\mathbf{x}(t+dt) = A_d\mathbf{x}(t) + B_d u(t) + \alpha\varepsilon_t B_d u(t) + \xi_t, \tag{3}$$

$$\mathbf{y}(t) = \mathbf{x}(t-\delta t) + \sigma_t \tag{4}$$

The matrices $A_d$ and $B_d$ form the discrete-time state space representation of the continuous-time system defined in *Equation 1*, which for a discretization step of $dt$ corresponds to: $A_d = e^{dtA}$, and $B_d = \left(\int_0^{dt} e^{sA} ds\right) B$. The constant $\alpha > 0$ is a scaling parameter; $\varepsilon_t$, $\xi_t$ and $\sigma_t$ are Gaussian noise disturbances. The multiplicative noise ($\varepsilon_t$) is a scalar with zero mean and unit variance, whereas the additive sources of noise are 4-dimensional random disturbances with zero mean and variance set to $\Sigma_{\xi,\sigma}$, which will be defined below. The measurement delay was $\delta t = 100 ms$ in a agreement with measured and modeled latencies of rapid saccadic responses to visual stimuli (*Munoz and Everling, 2004*; *Stanford et al., 2010*). The subscript $t$ for the random noise disturbances was used to remind that these series do not have finite instantaneous variation; but they have finite variance over the discretization interval of $dt$.

The variable $\mathbf{y}(t)$ defined in *Equation 4* represents the retinal information that is available to control the eye movement, which in the context of this paper is the state vector delayed by $\delta t$ and corrupted by the sensory ($\sigma_t$). Thus this definition captures the hypothesis that the available sensory information is a noisy and delayed measurement of the state. Recall that the augmented state vector includes position and velocity, as well as the target position and velocity.

## Closed-loop controller

Optimal estimation and control of the stochastic system defined in *Equations 3 and 4* can be derived in the framework of extended Linear-Quadratic-Gaussian control (LQG), including the effect of control and state-dependent noise (*Todorov, 2005*). However this approach is not necessarily well suited for handling sensorimotor delays because it requires system augmentation (*Crevecoeur and Scott, 2013*), and as a consequence the estimator achieves optimal (probabilistic) projection of the prior estimate onto the delayed state measurement (*Anderson and Moore, 1979*). Since we know that the visual system extrapolates sensory information to compute the present state of the eye or of a moving target (*Bennett et al., 2007*; *Ferrera and Barborica, 2010*; *Diaz et al., 2013*; *Blohm et al., 2005*), we were interested to derive an optimal estimator that explicitly extrapolates sensory signals, captured in $\mathbf{y}(t)$, over the interval $\delta t$ (see also *Figure 2*). The key aspect of this estimator design is that, by taking into account the control function $u(s)$, $t - \delta t \le s \le t$, the variance of the extrapolated sensory signal is dynamically adjusted as a function of the control-dependent noise (3$^{\text{rd}}$ term of *Equation 3*).

More precisely, we assume that neural processing of sensory signals consists in computing an estimate of the present state of the eye given the delayed sensory signals as follows:

$$\mathbf{x}(t|\mathbf{y}) = e^{\delta tA}\mathbf{y}(t) + \int_{t-\delta t}^{t} e^{(t-s)A} Bu(s)ds. \tag{5}$$

Using the notation $M(t) := e^{tA}$, it is easy to observe that the extrapolation error ($\Delta_t$) follows a Gaussian distribution defined as follows:

$$\mathbf{x}(t|\mathbf{y}) = \mathbf{x}(t) + \Delta_t, \tag{6}$$

$$\Delta_t \sim N(0, V_t), \tag{7}$$

$$V_t = M(\delta t)\Sigma_\sigma M(\delta t)^T + \int_{t-\delta t}^{t} \alpha^{'2} M(t-s)Bu(s)u(s)^T B^T M(t-s)^T ds, \tag{8}$$

where $\alpha' := \alpha(dt)^{-1/2}$ was defined in agreement with the unit-variance Brownian noise disturbance considered for the stochastic differential equation.

With these definitions, we can derive an adaptive estimator based on standard Kalman filtering using the extrapolated state (*Equation 8*) instead of the available state measurement (*Equation 4*). The state estimate is computed in two steps as follows:

$$\hat{\mathbf{x}}^P(t+dt) = A_d\hat{\mathbf{x}}(t) + B_d u(t) \tag{9}$$

$$\hat{\mathbf{x}}(t+dt) = \hat{\mathbf{x}}^P(t+dt) + K(t)(\mathbf{x}(t|\mathbf{y}) - \hat{\mathbf{x}}(t)), \tag{10}$$

and the Kalman gain, $K(t)$, as well as the covariance of the estimated state are updated iteratively following standard procedures (*Anderson and Moore, 1979*). Observe from *Equation 10* that a reduction in the Kalman gain has a direct impact on the use of visual feedback through the relationship between this feedback ($\mathbf{y}(t)$) and the sensory extrapolation ($\mathbf{x}(t|\mathbf{y})$, *Equation 5*). In other words, visual information conveyed in the sensory feedback participates less to the estimation when the Kalman gain is low.

Observe that the separation principle does not hold because the variances of the one-step prediction and of $\Delta_t$ both depend on $u_t$. Thus our approach is valid under the assumption that the control must not be jointly optimized with the state estimator. Instead of optimizing iteratively the controller and the state estimator as in the extended LQG framework (*Todorov, 2005*; *Phillis, 1985*), we computed the controller independently based on the heuristic assumption that the separation principle applied, and then optimized the state estimator defined in *Equations 9–10* by taking into account the effect of control-dependent noise explicitly (*Equations 6–8)*. The controller was thus obtained by solving the LQG control problem while ignoring the multiplicative noise in *Equation 3* as follows:

$$u(t) = -\left(R + B_d^T S_{t+1}B_d\right)^{-1} B_d^T S_{t+1}A\hat{\mathbf{x}}(t) \tag{11}$$

In *Equation 11*, $R$ represents the cost of motor commands, and the matrices $S_t$ are computed offline following standard procedures (*Todorov, 2005*; *Astrom, 1970*).

We developed this approach to include the extrapolation of sensory data while considering the control over the delay interval explicitly. Using feedback control based on a predicted state is known as *finite spectrum assignment* (FSA), which is germane to a Smith predictor in that it aims at removing the delay from the feedback loop (*Zhong, 2010*). Here, FSA was chosen to reconstruct the predicted state (instead of the system output as for the Smith predictor), allowing the use of position and velocity estimates in the control law (*Equation 11*).

## Numerical simulations

The only free parameters are $\alpha$ (the scaling of the signal dependent noise), the covariance matrices of $\xi_t$ and $\sigma_t$ (respectively $\Sigma_\xi$ and $\Sigma_\sigma$), and the cost-function used for control. We used the following values: the constant $\alpha$ was set to 0.08, $\Sigma_\xi$ was $10^{-3} \times B_d B_d^T$ and $\Sigma_\sigma$ was $10^{-6}$ times the identity matrix of appropriate dimension. These parameters were manually adjusted so that when adding the signal-dependent term to the variance of the extrapolation error (*Equation 8*), the Kalman gains converged to steady-state values and the variances of the extrapolated state and of the motor noise were comparable during fixation. It is clear that changing the noise parameters may influence the results qualitatively. However the key feature of the adaptive estimator is that the extrapolation variance increases monotonically with the square of the motor command, which is why the extrapolated measurement is dynamically reduced during movement. This aspect does not depend on the different noise parameters.

The cost parameters were adjusted to generate simulated saccades compatible with typical recordings of eye movements in humans, and these parameters do not impact the results qualitatively. For saccadic movements, we simulated two fixation periods at the initial ($x_1^*$) and final ($x_2^*$)

targets during which the cost of position error was $Q_{FIXATION, i} = (x_1 - x_i^*)^2$. The two fixation periods were separated by the movement time, which was a 50 ms window during which the eye was free to move without any penalty on the state vector. For the smooth movements in response to velocity jumps, we simulated a fixation to the target and changed the target state during a simulation run. The cost of motor commands in all cases was $Ru(t)^2$, with $R: = 0.01$. Finally we used a discretization step of 5 ms.

One difficulty is that the extrapolation requires that all state variables, including the target, be observed independently (*Equation 4*). This is not fully compatible with the visual system, because there is no measurement of the target state independent of the state of the eye. This limitation could be overcome by considering another observer that reconstructs the state vector prior to extrapolating the sensory feedback. Here, instead of considering such additional observer, we injected similar amounts of signal-dependent noise in the sensory feedback about the state of the eye as about the state of the target. This procedure was chosen for simplicity and captures the intuitive idea that if the eye position is very noisy, then information about the target location logically shares the same uncertainty.

## Acknowledgements

The authors would like to thank G Blohm for comments on an earlier version of this manuscript. FC is supported by a grant from F.R.S.-FNRS (grant number: 1.B.087.15F *Chargé de Recherches*, Belgium) and by grants of the Interuniversity Attraction Poles of the Belgian Science Policy (IPA Network on Dynamical Systems, Control and Optimization, DYSCO).

## Additional information

### Funding

| Funder | Grant reference number | Author |
| --- | --- | --- |
| Fonds De La Recherche Scientifique - FNRS | 1.B.087.15F | Frederic Crevecoeur |

The funders had no role in study design, data collection and interpretation, or the decision to submit the work for publication.

### Author contributions

FC, Conceptualization, Resources, Software, Formal analysis, Validation, Investigation, Visualization, Methodology, Writing—original draft, Writing—review and editing; KPK, Conceptualization, Formal analysis, Validation, Investigation, Visualization, Methodology, Writing—original draft, Writing—review and editing

### Author ORCIDs

Frédéric Crevecoeur, http://orcid.org/0000-0002-1147-1153

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
