## [Decision Letter]

Thank you for submitting your article "Saccadic suppression as a perceptual consequence of efficient sensorimotor estimation" for consideration by *eLife*. Your article has been reviewed by three peer reviewers, and the evaluation has been overseen by a Reviewing Editor and David Van Essen as the Senior Editor. The reviewers have opted to remain anonymous.

The reviewers have discussed the reviews with one another and the Reviewing Editor has drafted this decision to help you prepare a revised submission.

Summary:

The authors explore the hypothesis that saccadic suppression arises from efficient sensorimotor integration. That is, signals relating to eye position and speed may, and should, influence visual processing. This is a conceptually novel idea and an interesting new take on an old problem, as there is now an almost limitless volume of data showing that vision is attenuated during saccades and excellent evidence that this is generated through an internal mechanism. The problem is that we still struggle to understand how the attenuation comes about and why it appears to extend across such a large period of time (including long before saccades). The proposed implementation of the hypothesis as a well-defined control system adds to the rigor.

At the heart of the concept is the idea that the motor commands that are generated to make a saccade are interwoven into the neural network circuits that control perception. This is an appealing concept. The argument is that as the noise inherent in motor commands is amplitude-dependent, it is inevitable that uncertainty increases throughout the system when such commands are issued. Moreover, sensory feedback takes time, which is problematic during saccades as the sensory environment is clearly changing. The authors suggest that these two factors reduce the gain of the sensory input, leading to the reduced sensitivity observed during saccades. The idea of noise injection leading to saccadic suppression has been put forward previously but without the source being indicated. Given that the noise could be injected by the motor signals, it follows naturally that 'reduced sensory sensitivity' could follow.

Essential revisions:

All reviewers agreed that these are important, conceptually fresh ideas that may be potentially "game changing" in our understanding of a fundamental phenomenon (saccadic suppression). The reviewers thought, however, that the following revisions would strengthen the manuscript, making it more accessible to a wide audience and defining more clearly the experimental context that it falls in.

1) The manuscript needs a better description both regarding the text and, in particular, the math, which was deemed hard to follow. The description of the control model is very difficult to understand for a non-expert.

It was very difficult to link the equations to alterations in visual processing. For instance, take the essential parameter "y" in subsection “Closed-loop controller”. What is "y"? Does it represent visual signals, and if so, how do they 'calibrate' the sensory information relating to eye position and velocity? Is it somatonsensory information? If so, how does this tie together with saccade-related changes in sensory gain in the visual system? "y" is simply too general. The authors need to specify how "y" relates to the sensory codes that it represents, and explain how the equations relate back to modified visual processing.

While the model formulation seems to imply that the sensory feedback only represents 'eye position' the text refers to it more generally as 'eye state'. In control theory, state can refer to position as well as any of its derivatives (velocity, acceleration). The authors should be really clear on what they actually use, and also why their model only assumes position feedback. Would a model that uses both, position and velocity, make significantly different, maybe better predictions?

2) The work should be formulated more clearly as a hypothesis not a "result".

The step-by-step rationale with which the authors construct their main argument is clear, and the modeling work is sound. However, what is presented is less of a result than a very well-formulated hypothesis - that there is closed-loop visuo-motor feedback of saccadic control - which is expressed as a detailed quantitative model that provides specific and testable predictions.

It would also help if some over-statements were restructured or avoided. For instance, the last paragraph of the Introduction is confusing because it over-states the results and introduces ideas/concepts that only re-appear again in the Discussion section. It seems too strong a claim that the results are following from "phrasing this problem as a problem of shared resources.…". The presented model is not the result of formulating the problem in terms of shared resources. Rather the notion of shared resources is the authors' interpretation that follows from the modeling result and its nice match with reported perceptual suppression patterns in the literature. Similarly, the authors do not really "show theoretically that several features of saccadic suppression are expected.…" The one feature they theoretically predict is the time-window of suppression. The other predictions are more speculative and also qualitative.

The suggestion is to cut the entire last paragraph and move the notion of shared resources to the Discussion (together with Figure 1). It seems sufficient to state at the end of the Introduction that this study shows how an optimal control model that includes sensory feedback (closed-loop) can predict the observed perceptual and neural suppression pattern under the assumption of a common sensory pathway for perception and action.

3) The authors should put more effort into generating novel testable predictions of their hypothesis.

The authors state that the model is difficult to test experimentally. This may be true, but all reviewers agreed that it would greatly help if the authors included some (even if qualitative) novel experimental predictions. For instance, is there some relationship between motor output reliability and saccadic suppression that could be probed by training repeated saccades? Some effect of the reliability of the sensory input? Is there a reason why uncertainty in the sensory feedback signal due to image noise cannot in some way be traded-off with the motor noise induced by muscle activation?

For example, given that the close-loop assumption is essential, it would be nice to see predictions for various conditions that modulate signal uncertainty in said sensory feedback, and how that will affect perceptual suppression and movement accuracy. Similarly, since the authors suggest that the model can account for both saccadic and smooth eye movements, it would be nice to see a more nuanced discussion including predictions with regard to a gradual transition between the two movement modes, or more specifically, a gradual change in muscle activation that is the determining factor for kalman gain.

4) All reviewers found the magnocellular/parvocullular difference in suppression problematic. This distinction needs to be better explained, or alternatively, just dropped.

According to subsection “Saccadic Suppression as a Consequence of Optimal Estimation”: "This is interesting because changes in velocity typically have higher frequency contents than changes in position, thus a selective suppression of stimuli with low frequency contents may be directly related to the reduction in the weight of feedback about position, but not velocity." This statement is made in relation to the observation that low "spatial" frequency information (magnocellular) may be suppressed while high spatial frequency information (parvocellular) is not (or is suppressed less). Here it appears that the authors have generalized high frequency in the time domain to high frequency in the spatial domain. The temporal processing in the magnocellular and parvocellular systems actually overlap significantly. The authors need to carefully explain the relationship between space and time in their model and how this actually relates to visual processing (e.g. spatial frequency) before making major assertions about how their model may explain differences in suppression between parts of the visual system (e.g., how does this relate to the (absence of) suppression of color contrast?).

The results seem to predict that any component of the sensory feedback that is used for the prediction of eye position would be suppressed. In the Introduction, the authors state that saccadic suppression affects the entire magnocellular (where) pathway which includes motion (velocity), whereas the last paragraph of the Results explicitly excludes suppression of a velocity signal. Please, clarify this apparent contradiction.

5) The general point about the possible contribution of sensorimotor integration is worth making, but it needs to be balanced with a better discussion of the limitations of the hypothesis, and consideration of the full range of saccadic suppression phenomena.

The core experimental premise for the study's hypothesis is that there is closed-loop visuo-motor feedback of saccadic control. The empirical evidence for this is rather weak. The authors provide one relevant reference (Gaveau et al., 2003), (the West, Welsh and Pratt, 2009 reference does not appear to require sensory integration, only motor signal /CD integration). But, even in this study the influence of sensory input on saccade amplitude (~1 deg change for a 7 deg change in target location) is weak (and other studies typically report it to be zero). The authors should address how the magnitude of the sensory influence affects the control model. It is reasonable to use the Gaveau et al., 2003 paper as the seed for the model and the basis for hypothesis development, but the evidence is not definitive, and studies that found little evidence for closed-loop control should also be discussed, as an alternative viewpoint.

Related to this, Gaveau et al., 2003 claim that the sensory input is reflected in the motor signal within 50 ms. This may be un-physiologically fast. Because the authors lean heavily on the results of this paper, they should address how this number affects the outcome of the model.

With regard to perceived velocity and eye-movements, published results from Stephen Lisberger's group also seem relevant and should be discussed. Priebe and Lisberger (J Neurosci, 2004) found that initial smooth pursuit eye-movement velocities exhibit the same contrast-induced biases towards slow speeds as seen in perceived velocities of moving stimuli with eye fixation. These findings not only suggest that velocity might be part of the feedback signals but also that these feedback signals might already be the result of a Bayesian perceptual inference process. It would be useful if the authors could elaborate on this in the context of their model.

---

## [Author Response]

*Essential revisions:*

*All reviewers agreed that these are important, conceptually fresh ideas that may be potentially "game changing" in our understanding of a fundamental phenomenon (saccadic suppression). The reviewers thought, however, that the following revisions would strengthen the manuscript, making it more accessible to a wide audience and defining more clearly the experimental context that it falls in.*

*1) The manuscript needs a better description both regarding the text and, in particular, the math, which was deemed hard to follow. The description of the control model is very difficult to understand for a non-expert.*

*It was very difficult to link the equations to alterations in visual processing. For instance, take the essential parameter "y" in subsection “Closed-loop controller”. What is "y"? Does it represent visual signals, and if so, how do they 'calibrate' the sensory information relating to eye position and velocity? Is it somatonsensory information? If so, how does this tie together with saccade-related changes in sensory gain in the visual system? "y" is simply too general. The authors need to specify how "y" relates to the sensory codes that it represents, and explain how the equations relate back to modified visual processing.*

We have revised the Methods section to clarify this point. The vector y(t) is better defined and we provide more information about how the reduction in Kalman gain directly relates to the weight of sensory feedback. We also reworked the Results section to strengthen the link between the different concepts and the equations, and to give more intuition about how the predictions emerged from the model (subsection “Saccadic Suppression as a Consequence of Optimal Estimation”).

*While the model formulation seems to imply that the sensory feedback only represents 'eye position' the text refers to it more generally as 'eye state'. In control theory, state can refer to position as well as any of its derivatives (velocity, acceleration). The authors should be really clear on what they actually use, and also why their model only assumes position feedback. Would a model that uses both, position and velocity, make significantly different, maybe better predictions?*

We wish first to clarify that the model assumes position and velocity feedback as defined in Eqns. 1 and 4. This definition is now stated more clearly in the revised version of the manuscript (subsection “Biomechanical model”). We observed in the simulations that the weight of the velocity signals (second column of 𝐾 (𝑡)) does not display the same suppression as the weight of position signals (first column of (𝑡)), which is why we focused on the weight of position signals in the paper. A more careful description of the model predictions about velocity signals was also added to the revised Results (subsection “Saccadic Suppression as a Consequence of Optimal Estimation”)

*2) The work should be formulated more clearly as a hypothesis not a "result".*

*The step-by-step rationale with which the authors construct their main argument is clear, and the modeling work is sound. However, what is presented is less of a result than a very well-formulated hypothesis - that there is closed-loop visuo-motor feedback of saccadic control - which is expressed as a detailed quantitative model that provides specific and testable predictions.*

We made several changes to make it more apparent that it is a fully formulated hypothesis (Abstract; subsection “Saccadic Suppression as a Consequence of Optimal Estimation”; Discussion section). Clearly we do not present any experimental result, however our hypothesis emerges from non-trivial developments in the model (i.e. taking noise and delays into account explicitly), which is why we sometimes refer to it a theoretical result. We otherwise state more clearly that we suggest a novel hypothesis based on simulations.

*It would also help if some over-statements were restructured or avoided. For instance, the last paragraph of the Introduction is confusing because it over-states the results and introduces ideas/concepts that only re-appear again in the Discussion section. It seems too strong a claim that the results are following from "phrasing this problem as a problem of shared resources.…". The presented model is not the result of formulating the problem in terms of shared resources. Rather the notion of shared resources is the authors' interpretation that follows from the modeling result and its nice match with reported perceptual suppression patterns in the literature. Similarly, the authors do not really "show theoretically that several features of saccadic suppression are expected.…" The one feature they theoretically predict is the time-window of suppression. The other predictions are more speculative and also qualitative.*

*The suggestion is to cut the entire last paragraph and move the notion of shared resources to the Discussion (together with Figure 1). It seems sufficient to state at the end of the Introduction that this study shows how an optimal control model that includes sensory feedback (closed-loop) can predict the observed perceptual and neural suppression pattern under the assumption of a common sensory pathway for perception and action.*

We agree with the suggestion and made the following changes to address this point: we have moved the idea about shared resources in the Discussion and reworked the associated text as an interpretation of the model predictions (paragraph eight). We have also reworked Figure 4 (former Figure 1) to improve the link between our hypothesis about shared resources and the model shown in Figure 1 (former Figure 2).

*3) The authors should put more effort into generating novel testable predictions of their hypothesis.*

*The authors state that the model is difficult to test experimentally. This may be true, but all reviewers agreed that it would greatly help if the authors included some (even if qualitative) novel experimental predictions. For instance, is there some relationship between motor output reliability and saccadic suppression that could be probed by training repeated saccades? Some effect of the reliability of the sensory input? Is there a reason why uncertainty in the sensory feedback signal due to image noise cannot in some way be traded-off with the motor noise induced by muscle activation?*

Following this suggestion, we have added a paragraph about experimentally testable predictions that can be derived from the model. We suspect that the following effects may be uncovered: first it is conceivable that control can be impaired by forcing participants to rely more heavily on feedback than they should by presenting relevant stimuli during the interval of saccadic suppression. By increasing the reliance on feedback, the theory implies that saccade trajectories should become more variable across trials. As well, we mention that in principle, altering the reliability of sensory feedback should also modulate the magnitude of the saccadic suppression (Discussion, paragraph three).

*For example, given that the close-loop assumption is essential, it would be nice to see predictions for various conditions that modulate signal uncertainty in said sensory feedback, and how that will affect perceptual suppression and movement accuracy. Similarly, since the authors suggest that the model can account for both saccadic and smooth eye movements, it would be nice to see a more nuanced discussion including predictions with regard to a gradual transition between the two movement modes, or more specifically, a gradual change in muscle activation that is the determining factor for kalman gain.*

Regarding the transition between the two behaviors, we have clarified that this occurs as a result from differences related to correction for position error (which generates a saccade) or velocity. As the model is linear, the superposition principle applies and there is no switch in control between saccade and pursuit. Rather it is the correction for a position error that induces high control signals and generates the suppression. The simulated pursuit task uses smaller control signals that do not have visible impact on the Kalman gains. These points were added to the manuscript (paragraph four, subsection “Saccadic Suppression as a Consequence of Optimal Estimation”).

*4) All reviewers found the magnocellular/parvocullular difference in suppression problematic. This distinction needs to be better explained, or alternatively, just dropped.*

*According to subsection “Saccadic Suppression as a Consequence of Optimal Estimation”: "This is interesting because changes in velocity typically have higher frequency contents than changes in position, thus a selective suppression of stimuli with low frequency contents may be directly related to the reduction in the weight of feedback about position, but not velocity." This statement is made in relation to the observation that low "spatial" frequency information (magnocellular) may be suppressed while high spatial frequency information (parvocellular) is not (or is suppressed less). Here it appears that the authors have generalized high frequency in the time domain to high frequency in the spatial domain. The temporal processing in the magnocellular and parvocellular systems actually overlap significantly. The authors need to carefully explain the relationship between space and time in their model and how this actually relates to visual processing (e.g. spatial frequency) before making major assertions about how their model may explain differences in suppression between parts of the visual system (e.g., how does this relate to the (absence of) suppression of color contrast?).*

We realize that this argument was not clearly presented, and generalized abusively the link between spatial and temporal frequency contents linked by the scanning speed. Our point was that the difference between the suppression of position and velocity signals may relate to the selective suppression of signals with specific frequency contents. However as this point is more speculative and was considered problematic we followed the suggestion and removed this argument from the revised paper.

*The results seem to predict that any component of the sensory feedback that is used for the prediction of eye position would be suppressed. In the Introduction, the authors state that saccadic suppression affects the entire magnocellular (where) pathway which includes motion (velocity), whereas the last paragraph of the Results explicitly excludes suppression of a velocity signal. Please, clarify this apparent contradiction.*

We have clarified the related paragraph to avoid the previously ambiguous statement (Introduction section, paragraph three).

*5) The general point about the possible contribution of sensorimotor integration is worth making, but it needs to be balanced with a better discussion of the limitations of the hypothesis, and consideration of the full range of saccadic suppression phenomena.*

*The core experimental premise for the study's hypothesis is that there is closed-loop visuo-motor feedback of saccadic control. The empirical evidence for this is rather weak. The authors provide one relevant reference (Gaveau et al., 2003), (the West, Welsh and Pratt, 2009 reference does not appear to require sensory integration, only motor signal /CD integration). But, even in this study the influence of sensory input on saccade amplitude (~1 deg change for a 7 deg change in target location) is weak (and other studies typically report it to be zero). The authors should address how the magnitude of the sensory influence affects the control model. It is reasonable to use the Gaveau et al., 2003 paper as the seed for the model and the basis for hypothesis development, but the evidence is not definitive, and studies that found little evidence for closed-loop control should also be discussed, as an alternative viewpoint.*

We have rephrased the discussion to provide more support to the use of retinal feedback in a control model of saccades. More precisely, we acknowledge that although evidence is not definitive, and difficult to establish given that movements are fast and that delays are long, previous studies show that retinal information during or prior to saccades can be used to modify the eye trajectories, and that online corrections were elicited in the particular case of long saccades reported by Gaveau and colleagues (Discussion section, paragraph six). The observation that online corrections evoke only small adjustments (1deg for 7deg jump) is compatible with the model, because sensory information about the target jump is partially suppressed. As a consequence, the estimated target position is only partially corrected towards the new location and the online correction is only partial. We also added this argument to the revised Discussion.

*Related to this, Gaveau et al. claim that the sensory input is reflected in the motor signal within 50 ms. This may be un-physiologically fast. Because the authors lean heavily on the results of this paper, they should address how this number affects the outcome of the model.*

We have added to the manuscript how the value of the delay impacts the model predictions: we consider relatively long delays compatible with the latency of reflexive saccades (~100ms). Delays of ~50ms approach retinal processing times, which indeed seem extremely fast, however in this case the model still predicts that suppression occurs 50ms prior to movement onset. This information was added to the manuscript (paragraph three, subsection “Saccadic Suppression as a Consequence of Optimal Estimation”).

*With regard to perceived velocity and eye-movements, published results from Stephen Lisberger's group also seem relevant and should be discussed. Priebe and Lisberger (J Neurosci, 2004) found that initial smooth pursuit eye-movement velocities exhibit the same contrast-induced biases towards slow speeds as seen in perceived velocities of moving stimuli with eye fixation. These findings not only suggest that velocity might be part of the feedback signals but also that these feedback signals might already be the result of a Bayesian perceptual inference process. It would be useful if the authors could elaborate on this in the context of their model.*

We wish to thank the reviewers for bringing this reference to our attention. As mentioned above, we have clarified that the control model considers a state vector that includes velocity, thus this signal is also used in the controller. We have added the suggested reference in the manuscript as it also highlights overlap between perceptual and motor system in the context of smooth pursuit (Discussion section, paragraph eight). In the same paragraph we added a recent paper by Chen and colleagues who show a strong link between saccadic suppression and motor layers in superior colliculus, which is clearly relevant to our study.